# shRNAs Targeting a Common *KCNQ1* Variant Could Alleviate Long-QT1 Disease Severity by Inhibiting a Mutant Allele

**DOI:** 10.3390/ijms23074053

**Published:** 2022-04-06

**Authors:** Lucía Cócera-Ortega, Ronald Wilders, Selina C. Kamps, Benedetta Fabrizi, Irit Huber, Ingeborg van der Made, Anouk van den Bout, Dylan K. de Vries, Lior Gepstein, Arie O. Verkerk, Yigal M. Pinto, Anke J. Tijsen

**Affiliations:** 1Department of Experimental Cardiology, Amsterdam Cardiovascular Sciences, Amsterdam UMC, University of Amsterdam, Meibergdreef 9, 1105 AZ Amsterdam, The Netherlands; l.coceraortega@amsterdamumc.nl (L.C.-O.); selina.kamps@live.nl (S.C.K.); b.fabrizi@amsterdamumc.nl (B.F.); i.vandermade@amsterdamumc.nl (I.v.d.M.); anoukvdbout@gmail.com (A.v.d.B.); d.k.devries@amsterdamumc.nl (D.K.d.V.); a.o.verkerk@amsterdamumc.nl (A.O.V.); y.pinto@amsterdamumc.nl (Y.M.P.); 2Department of Medical Biology, Amsterdam Cardiovascular Sciences, Amsterdam UMC, University of Amsterdam, Meibergdreef 9, 1105 AZ Amsterdam, The Netherlands; r.wilders@amsterdamumc.nl; 3The Sohnis Family Laboratory for Cardiac Electrophysiology and Regenerative Medicine, Rappaport Faculty of Medicine and Research Institute, Technion—Israel Institute of Technology, Haifa 3109601, Israel; orr@technion.ac.il (I.H.); mdlior@technion.ac.il (L.G.)

**Keywords:** long-QT syndrome type 1, hiPSC-cardiomyocytes, RNA interference, gene therapy, arrhythmia

## Abstract

Long-QT syndrome type 1 (LQT1) is caused by mutations in *KCNQ1*. Patients heterozygous for such a mutation co-assemble both mutant and wild-type *KCNQ1*-encoded subunits into tetrameric Kv7.1 potassium channels. Here, we investigated whether allele-specific inhibition of mutant *KCNQ1* by targeting a common variant can shift the balance towards increased incorporation of the wild-type allele to alleviate the disease in human-induced pluripotent stem-cell-derived cardiomyocytes (hiPSC-CMs). We identified the single nucleotide polymorphisms (SNP) rs1057128 (G/A) in *KCNQ1*, with a heterozygosity of 27% in the European population. Next, we determined allele-specificity of short-hairpin RNAs (shRNAs) targeting either allele of this SNP in hiPSC-CMs that carry an LQT1 mutation. Our shRNAs downregulated 60% of the A allele and 40% of the G allele without affecting the non-targeted allele. Suppression of the mutant *KCNQ1* allele by 60% decreased the occurrence of arrhythmic events in hiPSC-CMs measured by a voltage-sensitive reporter, while suppression of the wild-type allele increased the occurrence of arrhythmic events. Furthermore, computer simulations based on another LQT1 mutation revealed that 60% suppression of the mutant *KCNQ1* allele shortens the prolonged action potential in an adult cardiomyocyte model. We conclude that allele-specific inhibition of a mutant *KCNQ1* allele by targeting a common variant may alleviate the disease. This novel approach avoids the need to design shRNAs to target every single mutation and opens up the exciting possibility of treating multiple LQT1-causing mutations with only two shRNAs.

## 1. Introduction

Congenital long-QT syndrome (LQTS) is the most common cardiac channelopathy with a prevalence of 1:2500 healthy live births [1]. LQTS is characterized by a prolonged ventricular action potential duration (APD) at the cellular level and a prolonged QTc interval on the electrocardiogram (ECG). LQTS can result in life-threatening arrhythmias as a result of impaired ventricular repolarization. 

Congenital LQTS is caused by mutations in genes encoding ion channel proteins and membrane adaptor proteins. Pathogenic mutations in *KCNQ1*, *KCNH2*, and *SCN5A* cause long-QT syndrome types 1, 2, and 3 (LQT1, LQT2, and LQT3), respectively, and together account for 97% of patients with genetically confirmed LQTS [2]. The most prevalent form is LQT1, with an occurrence of 40–55% of genetically confirmed LQTS [3]. LQT1 is caused by mutations in the *KCNQ1*-encoded α-subunit of the Kv7.1 potassium channel, which is responsible for the slow component of the delayed rectifier current (I_Ks_) during the repolarization phase of the working myocardial action potential [4].

LQT1 is effectively treated with β-blockers, even in patients with a genetic diagnosis but normal QTc [5,6]. However, treatment with β-blockers does not completely prevent aborted cardiac arrest or sudden cardiac death, and especially patients with cardiac events before starting β-blocker therapy are still at a high risk of experiencing recurrent events while on β-blocker therapy [7]. Furthermore, β–blockers frequently come with side effects or are not tolerated at all, which leads to refractoriness and non-compliance with the therapy [8,9], and as a result, life-threatening arrhythmias may still occur in these patients. When β-blocker medication fails or is ill tolerated, only invasive measures, such as the implantation of cardioverter-defibrillators and/or left cardiac sympathetic denervation, remain [10]. Overall, these LQT1 treatments fail to address and treat the root of the disease, which is the presence of a mutant KCNQ1 protein. Together, this highlights the need for other, more effective therapeutic approaches, which may be found in directly targeting the mutant protein itself. 

Since *KCNQ1*-encoded α-subunits post-translationally tetramerize to jointly form one functional Kv7.1 channel, patients heterozygous for LQT1-causing mutations assemble Kv7.1-channels with both wild-type and mutant subunits, where many LQT1-causing mutations display some degree of a dominant-negative effect [11,12,13]. Our group showed previously that the balance between the expression of wild-type and mutant *KCNQ1* alleles affects disease severity, even between family members carrying the same LQT1-causing mutation [14]. In this study, patients with repressive single nucleotide polymorphisms (SNPs) in the 3′-untranslated region (3′UTR) of the mutant *KCNQ1* mRNA had a shorter QTc and fewer symptoms. This suggests that intentional repression of the mutant *KCNQ1* allele, e.g., by RNA interference, might alleviate the disease and decrease arrhythmias. Selective targeting of mutant alleles with allele-specific small interfering RNAs (siRNAs) has indeed been shown to be a promising therapeutic approach for other cardiac syndromes [15,16,17,18,19]. In this regard, allele-specific downregulation of *RYR2*- and *KCNH2*-encoded mutant alleles that cause catecholaminergic polymorphic ventricular tachycardia type 1 (CPVT1) and LQT2, respectively, successfully decreased the occurrence of arrhythmic events in mice and human-induced pluripotent stem-cell-derived cardiomyocytes (hiPSC-CMs) [17,19]. However, in these studies, allele-specific siRNAs were designed to target specific causal mutations. While effective, this induces a major drawback. Since more than 600 mutations associated with LQTS have been identified in *KCNQ1* [3], such mutation-specific strategies will require the engineering of hundreds of allele-specific siRNAs to treat all patients. A dual-component gene therapy system combining complete mutation-independent *KCNQ1* suppression and replacement with wild-type *KCNQ1* that is not sensitive to the suppression could overcome this problem. This system shortens APD in LQT1 hiPSC-CMs to levels comparable to their isogenic control hiPSC-CMs [20]. However, fine-tuning this suppression–replacement *KCNQ1* therapy might be challenging and could result in undesired effects, such as excessive repression with insufficient replacement or inhomogeneous expression leading to repolarization heterogeneity.

In the present study we developed a versatile system using short-hairpin RNAs (shRNAs) to selectively silence the mutant *KCNQ1* allele targeting a common SNP in the same *KCNQ1* mRNA. By targeting a common SNP, we avoid making mutation-specific shRNAs and instead can generate only two allele-specific shRNAs that allow the treatment of all patients heterozygous for this SNP, irrespective of the actual disease-causing mutation carried by these patients. This strategy only requires determining which variant of the SNP resides on the mutant *KCNQ1* allele. We designed and validated allele-specific shRNAs to selectively target either allele of this common SNP in hiPSC-CMs carrying a LQT1 mutation. We show that specific inhibition of the mutant allele by these allele-specific shRNAs decreased the occurrence of arrhythmic events in hiPSC-CMs, while specific inhibition of the wild-type allele increased the occurrence of such events, demonstrating the functionality of the shRNAs targeting both alleles of this common SNP. 

## 2. Results

### 2.1. Allele-Specific Targeting of Common SNPs in KCNQ1

To allow allele-specific downregulation of *KCNQ1* by targeting common SNPs, we searched the coding region of *KCNQ1* for synonymous SNPs because these were expected to have the highest minor allele frequencies (MAF). Consequently, the highest number of patients will be heterozygous for these SNPs, which allows allele-specific targeting. We found 2 synonymous SNPs in *KCNQ1* with a MAF in the Genome Aggregation Database (gnomAD) population of 1.6% (rs17215465) and 16.6% (rs1057128), which translates to a heterozygosity of 3% and 27%, respectively. We selected rs1057128 in exon 13 for allele-specific targeting, as this SNP would allow the treatment of 27% of the patients, independent of the causal mutation. 

Because both alleles of the SNP can reside on the mutant allele in different patients, we designed shRNAs to target both alleles of rs1057128. We selected shRNAs targeting the A allele with a mismatch to the non-targeted G allele at positions 10, 13, 15, 16, and 18 and shRNAs targeting the G allele with a mismatch to the non-targeted A allele at positions 10, 11, 12, 16, and 18 (Figure 1). This selection was based on the results of Huang et al. [21], who provided insight on the mismatch positions with the highest discrimination potential, depending on the type of nucleotide changes. 

To test these shRNAs, we used two hiPSC-CM lines, Line 1 (clone 72) and Line 2 (clone 5K) (Appendix A), derived from two brothers with a pathogenic LQT1 mutation (R243C) in exon 5 of *KCNQ1*, which segregates in their extended family (Appendix A of [14]). These brothers were 42 and 40 years old, respectively, when dermal fibroblasts were obtained. Both brothers were asymptomatic and had a QTc interval at rest of 434 ms (age 36) and 509 ms (age 34), respectively. Furthermore, both brothers were heterozygous for SNP rs1057128 and carried the A allele on the mutant *KCNQ1* allele. 

The R243C mutation is located in the voltage-sensing S4 transmembrane domain of the Kv7.1 α-subunit and has been characterized with a slower current activation and a shift in the voltage dependency of both I_Ks_ activation and inactivation towards more positive potentials than I_Ks_ of wild-type subunits. In addition, while co-expression of *KCNQ1*-encoded α-subunits with *KCNE1*-encoded minK β-subunits induced a stronger current than expression of *KCNQ1*-encoded α-subunits alone, this current was strongly suppressed to non-functional levels when minK was co-expressed with R243C mutated α-subunits [22]. Furthermore, when the R243C mutant subunits form heterotetramers with wild-type subunits, channel activation by PKA and PKC is blunted, which highlights the dominant-negative nature of this mutation [23,24].

We lentivirally transduced hiPSC-CMs of both lines with the shRNAs targeting the A or G allele of SNP rs1057128 and compared by allele-specific quantitative real-time PCR (qRT-PCR) expression levels of mutant and wild-type *KCNQ1* alleles to expression levels in cells transduced with a scrambled negative control shRNA (shSCR). Targeting the A allele of rs1057128, which resides in these hiPSC lines on the mutant *KCNQ1* allele, revealed one shRNA, shA18, which downregulated the mutant *KCNQ1* allele without affecting the wild-type allele in both hiPSC lines (Figure 1a). ShA10 also downregulated the mutant allele in Line 1; yet, it was not allele-specific, and expression of the wild-type allele was partially lost. Previous studies suggested that introduction of additional mismatches in the seed region of shRNAs can improve discrimination between targeted and non-targeted alleles. Furthermore, decreased shRNA duplex stability at the 5′-part of the antisense strand in the so-called fork shRNAs increased inclusion of the functional antisense strand in the RNA-induced silencing complex (RISC), which improved allele-specific downregulation [25]. We introduced such additional mismatches to improve the allele-specific downregulation. We only introduced the additional mismatches to shA18 since shA10 strongly induced an interferon response (Appendix A), a possible response previously described upon introduction of foreign RNA [26]. Unfortunately, none of these modifications improved allele-specific downregulation of the mutant *KCNQ1* allele by targeting the A allele of rs1057128 (Appendix A). 

Targeting the G allele of SNP rs1057128, which resides in these hiPSC lines on the wild-type *KCNQ1* allele, revealed that 3 shRNAs, i.e., shG11, shG12, and shG18, downregulated in both hiPSC lines the wild-type *KCNQ1* allele without affecting the mutant allele (Figure 1b). The two other shRNAs, shG10 and shG16, were not allele-specific and also downregulated expression of the mutant *KCNQ1* allele. Because shG11, shG12, and shG18 did not trigger an interferon response (Appendix A), we introduced additional mismatches and created fork shRNAs for these shRNAs. However, also for these shRNAs targeting the G allele of SNP rs1057128, these modifications did not improve allele-specificity (Appendix A). 

### 2.2. Allele-Specific Targeting of SNPs in the 3′UTR of KCNQ1

In our search for common SNPs in the coding region of *KCNQ1*, we noticed several SNPs with a high MAF in the 3′UTR of *KCNQ1*. One of these SNPs is rs8234, with a MAF of 38.5% in the gnomAD population and a corresponding heterozygosity of 47%. This means that targeting this SNP would further increase applicability of allele-specific shRNAs to 47% of the patients. The younger brother described above (hiPSC Line 2) is also heterozygous for this SNP, and he carries the A allele on the mutant *KCNQ1* allele. Therefore, we tested in this line whether this SNP outside the coding region of *KCNQ1* also allowed allele-specific targeting.

We designed 7 shRNAs to target the A allele of rs8234. Unfortunately, although all were able to downregulate expression of the mutant *KCNQ1* allele, none was allele-specific (Appendix A). We also designed eight shRNAs to target the G allele of rs8234 on the wild-type *KCNQ1* allele. These shRNAs either did not induce a downregulation of any allele or, although not significantly, downregulated both alleles (Appendix A).

### 2.3. Allele-Specific shRNAs Affect the Allelic Balance in hiPSC-CMs

Mutations in *KCNQ1* often act as dominant-negative mutations [12]. Kv7.1 channels are formed by the assembly of four *KCNQ1*-encoded Kv7.1 α-subunits, and patients heterozygous for an LQT1-causing mutation combine wild-type and mutant subunits in their channels. The balance between the expression of wild-type and mutant subunits will determine the proportion of mutated subunits in the Kv7.1 channels and thus also the number of fully functional channels formed solely by wild-type subunits. Therefore, we assessed the extent to which the allele-specific downregulation by shRNAs affected the balance between wild-type and mutant allele expression in the hiPSC-CMs of our two lines. 

As discussed above, two of our shRNAs targeting the A allele of rs1057128 on the mutant *KCNQ1* allele downregulated the mutant allele in Line 1. Of these two, shA18 was allele-specific, while shA10 downregulated both mutant and wild-type alleles. In Line 2, only shA18 downregulated the mutant allele significantly. In Line 1, shA18 shifted the allelic balance from wild-type:mutant 43:57% to 63:37% (Figure 2a; left). Strikingly, despite the fact that shA10 was not allele-specific in Line 2, this shRNA induced the largest shift in allelic balance in this line as it shifted the wild-type:mutant ratio from 37:63% to 46:54% (Figure 2a; right), while shA18, which was allele-specific, shifted the balance not-significantly to 43:57%. These shifts in allelic imbalance in both lines, together with the induction of an interferon response by shA10 (Appendix A), caused us to select shA18 for further functional experiments. 

Of our shRNAs targeting the G allele of rs1057128 on the wild-type *KCNQ1* allele, three shRNAs were able to allele-specifically downregulate the wild-type *KCNQ1* allele. In line 1, these shRNAs shifted the wild-type:mutant ratio from 45:55% to 41:59% for shG11, 38:62% for shG12, and 39:61% for G18 (Figure 2b). In line 2, these shRNAs shifted the wild-type:mutant ratio from 59:41% to 44:56% for shG11, 44:56% for shG12, and 45:55% for G18 (Figure 2b). We selected shG11 for further functional experiments based on these shifts in allelic balance combined with the observation that shG11 was the most allele-specific of these three shRNAs (Figure 1b).

### 2.4. Specific Downregulation of the Mutant KCNQ1 Allele Prevents the Occurrence of Arrhythmic Events

To evaluate the functional effects of shA18 and shG11 in hiPSC-CMs, we used the fluorescence voltage indicator ArcLight A242. ArcLight is a voltage-sensitive, fluorescent protein that changes its conformation and thereby its fluorescence level in prompt response to the voltage dynamics of the action potential. ArcLight-expressing hiPSC-CMs show a reduction in fluorescence intensity in response to the depolarization of the cell, whereas the fluorescence intensity is restored in response to the subsequent repolarization [27]. We lentivirally transduced hiPSC-CMs with the shRNAs and a dsRED marker to allow for the selection of transduced cells. We recorded optical action potentials as fluorescence changes in ArcLight while cells were paced at 1 or 2 Hz to eliminate spontaneous beating and the associated effects of beating rate on APD. Because LQT1-causing mutations often act via dominant-negative mechanisms, which reduce I_Ks_ and compromise repolarization resulting in a prolongation of the APD, we expected that the APD of our LQT1 hiPSC-CMs would be shortened by the inhibition of the mutant *KCNQ1* allele as this would result in more fully wild-type functional tetramers. Surprisingly, we did not find a shortening of the APD at 50 or 80% of repolarization when the mutant *KCNQ1* allele was downregulated by shA18, but we actually detected a prolongation of APD_80_ both at 1 and 2 Hz (Figure 3, Appendix A). Downregulation of the wild-type *KCNQ1* allele by shG11, on the other hand, shortened APD_80_ at 1 Hz, and APD_20_ and APD_50_ at 2 Hz stimulation (Figure 3, Appendix A). These results are in contrast to what we expected from a shift towards less expression of the wild-type *KCNQ1* allele, where less expression of the wild-type allele would further decrease the amount of fully functional Kv7.1 channels and therefore decrease the repolarizing I_Ks_. These results might indicate that the loss of functional I_Ks_ also affects other ion channels and thereby the electrophysiological characteristics of hiPSC-CMs with a reduced functional I_Ks_.

QTc is often used to diagnose LQTS. However, QTc duration in LQT1 patients is not always directly related to life-threatening arrhythmias [14], and these arrhythmias may also occur in patients with a marginally prolonged QTc, although to a much lesser extent [28]. This means that our shRNAs, although they surprisingly did not affect the APD as expected, still might affect the occurrence of arrhythmic events. Therefore, we evaluated the occurrence of arrhythmic events in shRNA-treated hiPSC-CMs. Irregular action potential generation and/or membrane depolarizations were classified as arrhythmic events during spontaneous activity or while pacing (Figure 4a). In cells treated with the negative control shRNA, we detected arrhythmic events in 7% and 12% of the hiPSC-CMs of Lines 1 and 2, respectively. Downregulation of the mutant *KCNQ1* allele by shA18 abolished the occurrence of arrhythmic events in Line 1 and reduced them to 9% in Line 2 (Figure 4, *p* = 0.234 and *p* = 0.622 compared to shSCR negative control, respectively (chi-square test)). This indicates that the reduction of the mutant *KCNQ1* allele and the resulting shift in allelic imbalance towards the wild-type *KCNQ1* allele might improve the LQT1 phenotype even though no effect on APD was observed. Further in line with these findings, downregulation of the wild-type *KCNQ1* allele by shG11 increased the number of hiPSC-CMs with arrhythmic events to 56% in Line 1 and 51% in Line 2 (Figure 4, *p* < 0.001 compared to shSCR negative control for both lines (chi-square test)).

### 2.5. Computer Simulations Demonstrate the Applicability of Allele-Specific Inhibition in an Adult Human Cardiomyocyte Model

To investigate the applicability of the allele-specific shRNAs in treating LQT1 patients beyond the effects shown for the R243C mutation present in our hiPSC lines, we performed computer simulations in an adult human cardiomyocyte model. In this model, we first assumed that the presence of a single mutant subunit in the tetrameric channel fully abrogates the channel function. Because both shRNAs targeting the A and G allele of rs1057128 downregulate the targeted allele by 40–60%, we investigated the effect of a 60% reduction of mutant channels. In a situation where both mutant and wild-type *KCNQ1* alleles are equally expressed and co-assemble randomly, only 1 out of 16 of the I_Ks_ channels (6.25%) will consist of only wild-type subunits. Reduction of the mutant *KCNQ1* allele by 60% would increase this number to 26% (Figure 5a). In a situation where only I_Ks_ channels entirely built of wild-type subunits are conductive, this might increase I_Ks_ by approximately four times. This was indeed the case (Appendix A) and was accompanied by a 30% decrease in APD_90_ in both the epicardial and endocardial simulations and a 12% decrease in the mid-myocardial ones (Appendix A).

In a second simulation experiment, we built on experimental data by Vanoye et al. [29] to include characteristics of heterotetrameric channels formed by mutant subunits carrying an E160K mutation in KCNQ1 and wild-type subunits. Again, we simulated what would be the result of a 60% reduction in the number of mutant subunits. Restoration of I_Ks_ through a 60% suppression of the E160K mutant subunit expression is not very effective if I_Ks_ channels with 1-4 mutant subunits contribute equally to the heterotetrameric I_Ks_, which was simulated by a 68.4% reduction in I_Ks_ conductance and a +7.9 mV shift in the I_Ks_ steady-state activation curve as observed by Vanoye et al. [29]. This is due to the fact that the beneficial effects of the increased fraction of channels with only wild-type subunits is largely counteracted by the reduced total number of channels due to the reduced expression of subunits (Figure 5b,c; simulations labelled ‘Suppression 1’). However, if only channels with a single mutant KCNQ1 subunit are conductive and channels with 2–4 mutant subunits are not, 60% suppression of E160K mutant subunits is effective (Figure 5b,c; simulations labelled ‘Suppression 2’) because the fraction of channels with only one mutant subunit increases substantially (Figure 5a). This assumption seems reasonable for the E160K mutation, considering that the peak amplitude of the heterozygous WT/E160K I_Ks_ measured by Vanoye et al. [29] was 31.6 ± 5.9% (mean ± SEM, *n* = 34) of the homozygous wild-type control and the fraction of I_Ks_ channels with three or four wild-type subunits (so, maximally one mutant subunit) accounts for 31% of the channels (25% and 6%, respectively) when both alleles are equally expressed (Figure 5a).

## 3. Discussion

In this study we developed a versatile system to selectively silence an LQT1-causing mutant allele by targeting a common SNP in the *KCNQ1* gene. By targeting a common SNP, we avoided the need to make shRNAs against every single mutation, and we can apply our developed shRNAs independent of the actual disease-causing mutation as long as the patient is heterozygous for this SNP. This only requires determining which variant of the SNP resides on the mutant *KCNQ1* allele. Specifically, we designed allele-specific shRNAs to selectively target SNP rs1057128 in *KCNQ1* with a MAF of 16.6% and a corresponding heterozygosity of 27%. We validated these shRNAs in hiPSC-CMs from two LQT1 patients carrying an R243C mutation in *KCNQ1*, and we achieved a downregulation of the targeted allele of up to 60%. Furthermore, we showed that this specific inhibition of the mutant *KCNQ1* allele decreased the occurrence of arrhythmic events in the hiPSC-CMs, while inhibition of the wild-type allele had, as expected, the opposite effect and increased the occurrence of arrhythmic events. This underlines that our approach allows for intentionally shifting the balance between the targeted and non-targeted allele as desired, which might improve disease severity in patients. In addition, computer simulations assuming a heterozygous E160K mutation in *KCNQ1* show that a 60% reduction of the mutant *KCNQ1* allele may substantially increase the remaining I_Ks_ and shorten the prolonged APD. Together, these data indicate the applicability of our allele-specific shRNAs to act independently of the individual LQT1-causing mutation.

Patients heterozygous for a dominant-negative LQT1-causing mutation combine the translated products from normal and mutant alleles to form tetrameric channels. When both alleles are expressed at similar levels, only 6.25% of the channels will consist of purely wild-type KCNQ1 subunits. When the mutant *KCNQ1* allele is downregulated, the percentage of channels consisting purely of wild-type KCNQ1 subunits will increase (Figure 5a), which improves the repolarization capacity of the cells. This is supported by our previous study [14], where we showed that suppressive SNPs on the mutant *KCNQ1* allele reduce the prolonged QTc duration and the occurrence of symptoms. This indicates that a shift in allelic balance towards less expression of a particular LQT1-causing mutant KCNQ1 protein prevents the arrhythmic substrate and life-threatening events [14]. Generally, common SNPs have a small effect on the occurrence of diseases. However, in our previous study [14], a small effect caused by common SNPs was enough to explain part of the LQT1 disease variability and to protect mutation carriers against life-threatening events. This suggests that a stronger downregulation of the disease-causing allele could result in stronger protection against the occurrence of these life-threatening arrhythmias. Indeed, we showed that allele-specific silencing of the mutant *KCNQ1* allele, by shRNAs targeting rs1057128, by 40–60% in hiPSC-CMs shifted the allelic imbalance towards a lower expression of the mutant *KCNQ1* allele and decreased the occurrence of arrhythmic events in an LQT1 hiPSC-CM model. 

*KCNQ1* mutations are often classified in two groups based on the reduction of I_Ks_. Mutations in the first group resulted in mutant subunits that co-assembled with wild-type subunits, leading to heterotetrameric dysfunctional channels on the cell membrane. These mutations are classified as dominant-negative because they reduced the total I_Ks_ by more than 50%, which indicates that the mutant KCNQ1 protein interfered with the wild-type KCNQ1 protein [11]. Patients carrying such dominant-negative mutations would clearly benefit from a specific reduction in the amount of mutant KCNQ1 protein because this would increase the amount of fully functional homotetrameric channels formed solely by wild-type subunits. The second group of mutations are classified as causing haploinsufficiency because they reduced the total I_Ks_ up to 50% [30]. Underlying molecular mechanisms are defects in co-assembly or trafficking that prevent the mutant subunits from being transported to the cell membrane. In theory, these types of mutations would not benefit from the suppression of the mutant protein because they do not interfere with the function of the wild-type subunits. However, the trafficking of different cardiac channels is intertwined, and the trafficking disruption of one channel might affect other channels. For instance, mutations in *KCNQ1* that affect trafficking of Kv7.1 subunits have been found to also disrupt trafficking of the *KCNH2*-encoded hERG protein, causing a severe LQTS phenotype [31]. Interestingly, most of the loss-of-function mutations in *KCNQ1* exert some degree of trafficking defect [12], which implies that those mutations could also disrupt the trafficking of hERG. Therefore, a decrease in trafficking-impaired mutant Kv7.1 subunits by allele-specific shRNAs might prevent the sequestration of hERG channels by mutant Kv7.1 subunits and thus allow hERG to traffic to the cell membrane. This would increase repolarizing currents at the cell membrane and thus improve disease severity of LQTS, even in patients carrying a *KCNQ1* mutation leading to haploinsufficiency.

Previous studies with allele-specific siRNAs targeting specific mutations have shown that the complete silencing of mutant mRNA is not necessary to achieve therapeutic effects. In a mouse model of hypertrophic cardiomyopathy caused by an *Myh6* mutation, a reduction of 28.5% of mutant *Myh6* was enough to prevent the development of HCM [32]. Furthermore, in a mouse model of autosomal dominant centronuclear myopathy (AD-CNM), treatment with allele-specific siRNAs at advanced stages of the disease reduced around 40% of the mutant *DNM2* allele expression and partially rescued the muscle force impairment and morphologic abnormalities [33]. We found that a reduction of 60% of mutant *KCNQ1*, thus increasing the number of channels consisting of only wild-type subunits from 6.25% to 26%, was enough to substantially shorten the prolonged APD in an adult human cardiomyocyte computer model of an LQT1-causing mutation in *KCNQ1* (Figure 5, Appendix A). We also observed that a reduction of about 60% of the mutant *KCNQ1* allele was enough to decrease the number of arrhythmic events, while a reduction of about 40% of the wild-type *KCNQ1* allele was enough to elicit a high number of arrhythmic events in hiPSC-CMs carrying the R243C mutation in *KCNQ1.* These results suggest that an allele-specific reduction of 40% might be enough to functionally shift the allelic balance of wild-type and mutant *KCNQ1* alleles in LQT1.

Surprisingly, we only observed an effect of the allelic imbalance on the occurrence of arrhythmic events and not on the APD in our hiPSC-CMs. For this study, we used two hiPSC lines harboring the R243C mutation in *KCNQ1*, which might not be a very severe mutation [22,34]. Especially the patient from whom Line 1 was derived had only a borderline [6] prolonged QTc of 434 ms on the ECG at rest. The reduced severity of this particular mutation is further supported by the electrophysiological characterization of the R243C, W248R, and E261K mutations in *KCNQ1* in *Xenopus* oocytes by Franqueza et al. [22], who observed that the W248R and E261K mutations in the S4-S5 linker of *KCNQ1* reduced the generated current by 80% to 100% compared to wild-type KCNQ1 current, while the R243C mutated KCNQ1 only reduced this current by 60% [22]. Furthermore, hiPSC-CMs of both lines do not display a long APD compared to hiPSC-CMs of previously published LQT1 lines with different mutations [20,35,36]. The absence of an extremely long APD in the present study may be obscured due to the huge APD variability between hiPSC-CMs lines [37], but it could also be mutation-dependent [38] and a further indication of the reduced severity of this particular R243C mutation. In addition, hiPSC-CMs are characterized by immaturity, and they lack the inward rectifier potassium current (I_K1_), which results in a more depolarized resting membrane potential than adult cardiomyocytes [39]. In such a situation, a reduction in repolarizing currents can further depolarize the resting membrane potential, as has been found for rapid delayed rectifier potassium current (I_Kr_) reduction [40] and I_Ks_ blockade by 10 µM chromanol 293B [41], although the latter did not reach statistical significance due to the low number of cells measured. Further studies are required to determine the exact role of I_Ks_ in setting the resting membrane potential in hiPSC-CMs, but a depolarization of the resting membrane potential will have a strong impact on the activation of many ion currents [39], which could in turn give rise to unexpected changes in APD as the ones we observed in our hiPSC-CMs. Our computer simulations in an adult cardiomyocyte model with physiological resting membrane potential, which show the expected shortening of APD_90_ when the amount of dominant-negative protein is reduced by 60% as induced by our shRNA, further support that this immaturity might be the underlying reason for the unexpected APD changes in our hiPSC-CMs. However, in light of a potential therapy, this should be experimentally confirmed by treating adult cardiomyocytes with our shRNAs. 

Although the lack of APD changes in hiPSC-CMs could be explained by the immaturity of these cells, we may also have detected an effect on the occurrence of arrhythmic events as a result of the allelic imbalance and independent of the effect on APD. We observed in our previous study [14] that an allelic imbalance by suppressive 3′UTR SNPs affected the occurrence of symptoms even when the analysis was corrected for QTc duration, which indicates that the allelic imbalance still affects the occurrence of symptoms via other mechanisms independent of the QTc duration. Furthermore, the QTc duration is not always useful for predicting serious arrhythmic events in carriers of LQTS mutations [42]. This means that patients might still benefit from inducing an allelic imbalance by allele-specific shRNAs even though we do not detect shortening of the APD when we suppress the mutant *KCNQ1* allele.

We targeted a common SNP to specifically inhibit the mutant *KCNQ1* allele and thereby expand the applicability of our allele-specific shRNAs to a wider number of patients, independent of their specific LQT1-causing mutations. There have been more than 600 mutations described in *KCNQ1*, which are associated with either LQT1 or much less frequently short-QT syndrome type 2 (SQT2) [43]. It would be a tremendous task to design and optimize allele-specific shRNAs for every single mutation. Furthermore, it is questionable whether drug regulatory agencies would allow patient/family-specific shRNAs without large clinical trials, which would be impossible for patient-specific shRNAs. In addition, allele specificity is highly dependent on the sequence to be targeted, which could mean that some mutations will not be suitable at all for allele-specific targeting [44]. Targeting a common SNP instead of a specific mutation increases the number of patients that can be treated with a single shRNA [45,46] and thus overcomes the necessity of designing hundreds of shRNAs for LQT1. Specifically, SNP rs1057128 has a heterozygosity of 27%, which means that 27% of patients (LQT1 and SQT2) could potentially be treated with our allele specific-shRNAs. Therefore, we developed allele-specific shRNAs for each allele of the SNP (G/A), and in patients heterozygous for this SNP, it will only be necessary to determine which allele of the SNP resides on their mutant *KCNQ1* allele. In addition, since SQT2 is caused by gain of function mutations in *KCNQ1*, allele-specific shRNAs targeting common SNPs in *KCNQ1* could also be used to downregulate the mutant KCNQ1 protein in SQT2, further expanding the potential use of this system.

Targeting common SNPs instead of specific mutations opens up the possibility of treating a large number of diseases caused by dominant-negative mutations. It is estimated that a single individual carries about 4 million SNPs [47], the majority of which are common variants, accounting for 96–99% of the total variants present in a single individual [48]. Therefore, it is very likely that most of the genes in which dominant-negative mutations have been identified contain common SNPs. Our approach of targeting allele-specific shRNAs towards a common SNP could be applied to all those genes and alleviate the diseases that they cause. Dominant-negative mutations are also a common underlying disease mechanism in congenital cardiac diseases. As we described above, inhibition of a mutant protein rescued the phenotype in mice and/or hiPSCs-CMs with a mutation in *Myh6*, *RYR2*, and *KCNH2* [17,19,32], genes in which 47 to 978 different mutations are described and which contain common variants with a heterozygosity of at least 45%. This underlines the broad applicability of the approach of targeting a common SNP, as well as for other (cardiac) diseases where no evidence-based treatments currently exist, such as genetic cardiomyopathies caused by *LMNA* or *RBM20* mutations [49,50].

To allow for the transfer of these shRNAs to the clinic, several challenges are ahead of us. The effect of our shRNAs on APD in adult cardiomyocytes needs to be determined and explained if they also result in unexpected durations, as in the hiPSC-CMs. Afterwards, in vivo specificity and efficacy of the allele-specific shRNAs need to be shown in animal models. Furthermore, delivery methods and associated safety and dosage need to be addressed. In this study, we made use of a lentivirus as a delivery vector in hiPSC-CMs. However, the use of lentiviruses in cardiac gene therapy has several disadvantages since they present poor muscle transduction and integrate in the host genome, raising possible insertional mutagenesis [51]. Functional siRNA duplexes can be injected as naked siRNAs with or without chemical modifications for stability. However, they seem to mainly target the liver and are thus less efficient in targeting other organs [52]. Delivery of these siRNAs to the heart could be enhanced via nanoparticle delivery and/or conjugation to peptides. However, the effect of siRNAs will likely be lost over time, requiring repeated administration. Gene therapy delivery of shRNAs might be more efficient in targeting the human heart. The preferred vectors for the delivery of gene therapy to humans are the adeno-associated viruses (AAV) due to their safety profiles. Currently, there have been 51 clinical trials performed with proven efficacy using AAV vectors [53]. The advantage of our shRNAs of 45 bp in length is that their small transgene size ensures high AAV production titers. These high titers may be required to ensure that a high percentage of cells are hit, which is an especially important requirement for the treatment of LQT1. Targeting a low percentage of cardiomyocytes could result in heterogeneous cardiac repolarization between cells, which could also induce cardiac arrhythmias, particularly if this percentage varies throughout the ventricular tissue [54]. 

In conclusion, in this study we designed inhibitory RNAs to silence a disease-causing mutation in the *KCNQ1* gene while avoiding the need to generate siRNAs against every single mutation. We achieved this by targeting a common single nucleotide polymorphism, rs1057128, so that we were able to silence one allele specifically. When this allele contains the disease-causing mutation, computer simulations predict robust beneficial effects, such as the shortening of APD. Indeed, we demonstrated that specific inhibition of the mutant *KCNQ1* allele decreased the occurrence of arrhythmic events in hiPSC-CMs from two LQT1 patients. We envision that this approach might allow the development of two shRNAs that could improve the LQT1 phenotype in a substantial number of *KCNQ1* mutation carriers that are heterozygous for the targeted SNP.

## 4. Materials and Methods

### 4.1. Human iPSC Generation

All studies conform to the declaration of Helsinki and were approved by the Medical Ethics Committee of the Amsterdam UMC, Amsterdam. The skin biopsies were obtained after individual permission using standard informed consent procedures. Dermal fibroblasts were obtained from two brothers of 42 and 40 years of age with a diagnosis of familial LQT1 due to an R243C missense mutation in *KCNQ1*. Fibroblasts were retrovirally reprogrammed with the transcription factors OCT4, SOX2, and KLF4, with addition of valproic acid as described previously [55].

### 4.2. Human iPSC Culture

hiPSCs were cultured in mTeSR-1 (STEMCELL Technologies, Vancuver, BC, Canada, 85850) on plates coated with 1:500 diluted growth factor-reduced Matrigel (Corning, Bedford, MA, USA, FAL356231). Cells were passaged every 4–6 days via dissociation with 0.5 mM EDTA (Invitrogen, Grand Island, NY, USA, 15575-038) and seeded in mTeSR-1 supplemented with 2 μM Thiazovivin (Selleck Chemicals, Burlington, ON, Canada, S1459). Between passages, the mTeSR-1 medium was replaced every day, except for the first day after passaging. 

### 4.3. Karyotype Analysis

Karyotypes were determined using G-banding chromosome analysis according to standard procedures by the institutional cytogenetic laboratory of the Rambam Medical Center, Haifa, Israel.

### 4.4. Cardiac Differentiation of hiPSC

Differentiation towards cardiomyocytes was performed following a previously published protocol with slight adaptations [56]. Differentiation was induced 4 days after passaging by changing to CDM3 medium (RPMI-1640, Gibco, Paisley, UK 21875; 500 μg/ml human serum albumin, Sigma, St. Louis, MO, USA, A9731; 213 μg/ml l-ascorbic acid 2-phosphate, Sigma, St. Louis, MO, USA, A8960; 1% penicillin/streptomycin, Gibco, Grand Island, NY, USA, 15140-122) supplemented with 6 μM CHIR99021 (Stemgent; Beltsville, MD, USA, 04-0004-10) for two days, followed by CDM3 supplemented with 2 μM Wnt-C59 (Selleck Chemicals, Canada, S7037) for two days. From day 4 to day 10, the medium was changed every other day for the RPMI/B27 medium (RPMI-1640; 2% B27 supplement minus insulin, Gibco, Grand Island, NY, USA, A1895601; 1% penicillin/streptomycin). Spontaneous hiPSC-CM contractions could be identified from day 8 onwards. 

From day 10 onwards, metabolic cardiomyocyte selection was performed by replacing the hiPSC-CM medium once every week with CDM3 medium without glucose (RPMI-1640 without glucose, Gibco, Grand Island, NY, USA, 11879) supplemented with 20 mM sodium-lactate (Sigma-Aldrich, Switzerland L7022; dissolved in 1 M HEPES-solution) for at least 2 weeks [57]. After selection, the medium was replaced once a week with CDM3 medium with glucose. 

hiPSC-CM were either dissociated by TrypLE Express (Gibco, Grand Island, NY, USA, 12604) or TrypLE Select (Gibco, Grand Island, NY, USA, A1217701) with an incubation of 15 min and plated on Matrigel-coated plates or coverslips in RPMI/B27 medium with 2 μM Thiazovivin for shRNA selection experiments and basal characterization or in CDM3 medium containing lentivirus supplemented with 2 μM Thiazovivin for electrophysiology experiments. All experiments were conducted on hiPSC-CMs 40–60 days after the start of the differentiation, and each observation was replicated in 2 to 5 independent experiments with hiPSC-CMs from different differentiations. 

### 4.5. In Vitro Trilineage Differentiation Potential

Trilineage differentiation potential was assessed by the induction of endodermal, ectodermal, and mesodermal differentiation of hiPSCs using the STEMdiff Trilineage Differentiation Kit (STEMCELL Technologies, Vancuver, BC, Canada, 05230) according to the manufacturer’s protocol.

### 4.6. Immunocytochemistry

Cells for immunocytochemistry were plated on 12 mm, glass coverslips coated with Matrigel. Undifferentiated hiPSCs were cultured in mTeSR-1 for 3 days, and differentiated hiPSC-CMs were cultured for 1 week in RPMI/B27. Cells were fixed in 4% paraformaldehyde for 15 min at room temperature and washed 3 times in PBS. Cells were permeabilized with Triton X-100 in PBS for 8 min (0.1% for hiPSC-CMs and 1% for hiPSCs). Unspecific antibody binding was blocked by 20 min incubation with 4% goat or 10% horse serum. Primary antibodies (Appendix A) were diluted in PBS with 4% goat or 10% horse serum and incubated overnight at 4 °C. Cells were washed 3 times in PBST and incubated for 1 h at room temperature in the dark with 1:250 diluted secondary antibodies (Appendix A) in PBS with 4% goat or 10% horse serum. Cells were washed 3 times in PBST. Nuclei were counterstained with DAPI (1:5000) for 5 min and mounted in Mowiol (Sigma, St. Louis, MO, USA, 81381).

### 4.7. Plasmid Generation

For shRNA expression, we used the pLKO.1 backbone, either with puromycin as a selection marker for shRNA selection (pLKO.1-puro; Addgene, Teddington, UK, 8453) or with dsRED as a fluorescent marker for electrophysiology [58]. Cloning of shRNA sequences was similar in the pLKO.1-puro and pLKO.1-dsRED plasmids. Therefore, we designed the following oligonucleotides: forward 5′-CCGGAA-19 bp sense strand-TCAAGAC-19 bp antisense strand-TTTTTTTG-3′ and reverse 5′-AATTCAAAAAAA-19 bp sense strand-GTCTTGA-19 bp antisense strand-TT-3′. The sense strand is exactly the mRNA targeting sequence, and the antisense strand its reverse complementary sequence that will eventually bind the mRNA. We annealed 1 nmol of these oligonucleotides and cloned them into AgeI and EcoRI restriction sites in the pLKO.1 plasmids. Exact shRNA sequences are detailed in Appendix A. A shRNA with a scrambled sequence (shSCR) was used as a negative control shRNA [58]. pLV-CAG-ArcLight was previously described [59]. All plasmid sequences were verified by Sanger sequencing and the occurrence of mutations excluded.

### 4.8. Virus Production

To produce third-generation lentivirus of pLKO.1-puro, pLKO.1-dsRED- and pLV-CAG-ArcLight-based constructs, we co-transfected 4x∙10^6^ HEK293T cells with 4 μg of the expression plasmid, 2.7 μg pMDLg/pRRE, 1 μg pRSV-Rev, and 1.4 μg pVSVG using GeneJammer (Agilent, Cedar Creek, Tx, USA, 204130) according to the manufacturer’s protocol. The next day, the medium was replaced with CDM3 medium. This medium containing the produced lentivirus was collected after 24 h and either used directly for hiPSC-CM transduction or the number of transducing units (TU) was first determined. 

The amount of TU was determined by transducing 250,000 HEK293T cells with series of 50/100/200/500/1000 μL of medium with virus of the pLKO.1-dsRED plasmid. Three days after transduction, the cells were trypsinized and analyzed by FACS for the dsRED positive population. The condition with 10–20% positive cells was used to calculate the amount of TU, assuming 1 viral copy per cell. The amount of TU for an experimental virus and its corresponding control were determined in the same FACS experiment.

### 4.9. hiPSC-CMs Infection

For shRNA selection experiments, hiPSC-CMs were dissociated and replated in 6-well plates, 2 to 4 days before lentiviral transduction to ensure homogenous cell populations between conditions. For these selection experiments, 2 mL/well of medium with viruses containing the puromycin resistance cassette were freshly added to the hiPSC-CMs. Medium was refreshed the day after transduction. A total of 5 days after transduction, puromycin selection started with 8 μg/mL puromycin for 72 h, after which the cells were harvested for RNA experiments. 

For electrophysiology experiments, cells were dissociated and resuspended in 1.5 mL medium containing ArcLight-encoding lentivirus (TU not determined) and 30,000 TU of shRNA-encoding lentivirus and then plated in 35 mm optical plates (CELLview, Greiner Bio-one, Kremsmünster, Austria, 627860). Starting from day 2 onward, the medium was refreshed with CDM3 medium every other day. Cells were measured 6–8 days after plating and infection.

### 4.10. RNA Isolation

Total RNA was isolated using 1 mL TriReagent (Sigma-Aldrich; St. Louis, MO, USA, T9424). TriReagent was added directly to live cells growing on a dish. Total RNA isolation was performed according to the manufacturer’s protocol. 

### 4.11. qRT-PCR

To detect mRNA levels, 250 ng to 1 μg RNA was DNase-treated with DNase I amplification grade (Invitrogen, Carlsbad, CA, USA, 18068015) and reverse transcribed using Superscript II reverse transcriptase (Invitrogen, Carlsbad, CA, USA, 18064014) with oligo-dT and random hexamer primers according to the manufacturer’s protocol. cDNA was diluted 5 times, and 2 μL was used as input for qPCR. qPCR was performed using 1 μM primers (Appendix A) and LightCycler 480 SYBR Green master 1 (Roche, Mannheim, Germany, 04887352001) on a LightCycler 480 system II (Roche, Basel, Switzerland), using the following cycling program: 5 min pre-incubation at 95 °C; 40 cycles of 10 s denaturation at 95 °C, 20 s annealing (temperatures in Appendix A), and 20 s elongation at 72 °C. Data were analyzed using LinRegPCR quantitative PCR analysis software [60], and the starting concentration of transcripts estimated by this software was corrected for the geometric mean of 3 reference genes: HPRT, GAPDH, and TBP.

For allele-specific qRT-PCRs, allele specificity was obtained by allele-specific forward primers with the R243C mutation being the very last nucleotide on the 3′-end of the primer, which was combined with a common reverse primer. Allelic imbalance was assessed by comparing the expression of wild-type and mutant *KCNQ1* mRNA as percentages of the total *KCNQ1* expression, where the total *KCNQ1* is the sum of expression of both alleles. 

### 4.12. ArcLight Measurements

For ArcLight measurements, fluorescence was measured with a Leica TCS SP8 SMD mounted on a Leica DMI6000 inverted confocal microscope with a 40× oil inversion objective. ArcLight was excited with a 488 nm white light laser (WLL) pulsed with pulse picker and light collected with a 2HyD detector. For action potential recordings, fluorescence was recorded in XT line-scan mode with 512 pixels per frame at 1 frame per 2 ms. Only dsRed positive cells were measured in Tyrode’s solution containing (in mM): NaCl 140; KCl 5.4; CaCl_2_ 1.8; MgCl_2_ 1; HEPES 10; and glucose 10 (pH 7.4; NaOH), while incubated at 37 °C by an incubator enclosing the microscope. Cells were paced with field stimulation via carbon electrodes (P0003-7, EHT Technologies GmbH, Hamburg, Germany) at 1 or 2 Hz with a custom-made stimulator. 

XT recordings of fluorescence were converted into comma-separated value files with ImageJ for further analysis. Recordings were further analyzed by custom-made MATLAB software [59]. First, the fluorescence axis of the ArcLight optical signals was inverted. APD_20_, APD_50_, and APD_80_ were calculated as the median time interval of 5–12 action potentials required to reach 20, 50, and 80% of repolarization starting from 50% maximal upstroke height. For the comparison of arrhythmic events, each cell with an arrhythmic appearance, such as displaying irregular action potential generation and/or membrane depolarizations (examples in Figure 4) either at baseline while spontaneous beating or when stimulated, was counted as a cell with events. 

### 4.13. Computer Simulations

Functional effects of changes in I_Ks_ were assessed by computer simulations using the epicardial, midmyocardial, and endocardial versions of the human ventricular cell model by Ten Tusscher et al. [61], as updated by Ten Tusscher and Panfilov [62]. The numerical reconstruction was carried out on an Intel-i7-CPU-based workstation using Intel Visual Fortran and employing a simple and efficient Euler-type integration scheme with a timestep of 5 µs. Simulations were run for a sufficiently long time to achieve steady-state conditions.

### 4.14. Statistics

Data obtained from hiPSC-CMs are a combination of two to five independent experiments on cells from independent differentiations, with at least two biological replicates per experiment. Data of these independent experiments are combined using Factor Correction [63], where the control condition was used as a reference to calculate the correction factor by which all the data points of that experiment were corrected. As a consequence, data shown for continuous variables are a mean ± SEM of 6–15 biological replicates derived from 2–5 differentiations. For categorical data, the percentage of cells in all groups is depicted per condition. Continuous variables were analyzed with GraphPad Prism Software version 8, and the different groups were compared by a Kruskal–Wallis test in combination with Dunn’s post-hoc test. For comparisons of the allelic imbalance, the ratio between wild-type and mutant *KCNQ1* expression was used a continuous variable. Categorical variables were compared to the shSCR negative control by chi-square tests. *p* < 0.05 was considered significant.

## 5. Patents

L.C.-O., Y.M.P. and A.J.T. filed a patent application, owned by the Amsterdam UMC, that details claims related to the use of shRNAs targeting common variants for the treatment of long-QT syndrome type 1.

## Figures and Tables

**Figure 1 ijms-23-04053-f001:**
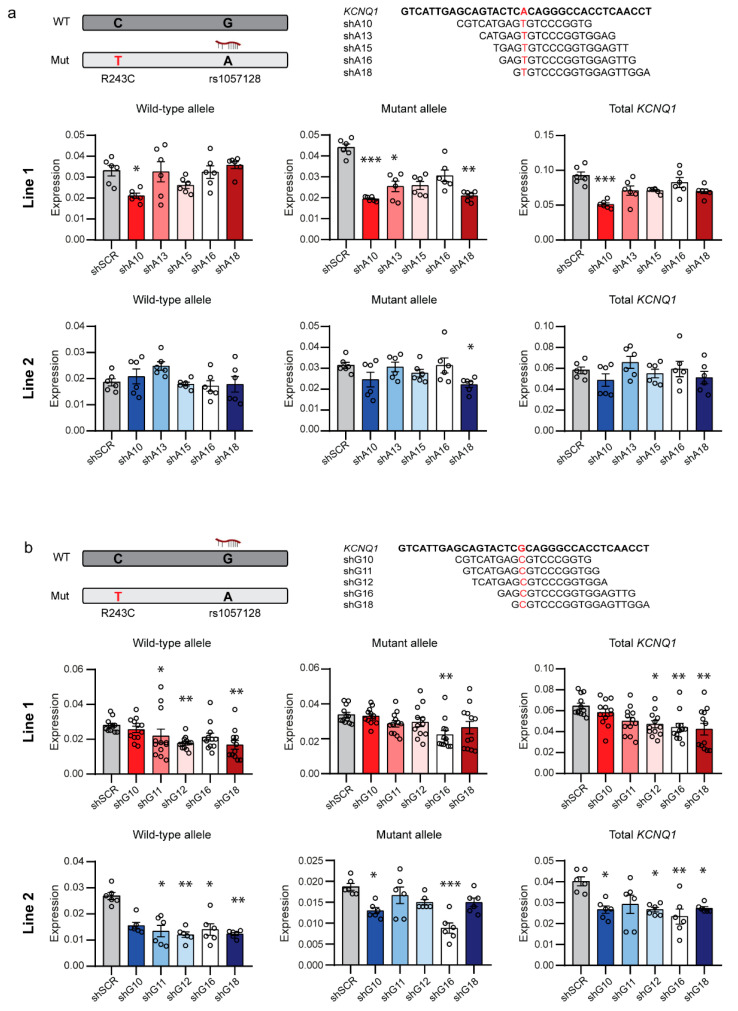
Allele-specific downregulation of *KCNQ1* expression in hiPSC-CMs by shRNAs targeting SNP rs1057128: (**a**) Top, schematic representation of the shRNAs targeting the A allele of rs1057128 on the mutant *KCNQ1* allele with the mismatch positions indicated in red. Middle, allele-specific relative mRNA expression of the wild-type and mutant allele and total *KCNQ1* in hiPSC-CMs of Line 1 (*n* = 6). Bottom, allele-specific relative mRNA expression of the wild-type and mutant allele and total *KCNQ1* in hiPSC-CMs of Line 2 (*n* = 6); (**b**) Top, schematic representation of the shRNAs targeting the G allele of SNP rs1057128 on the wild-type *KCNQ1* allele with the mismatch positions indicated in red. Middle, allele-specific relative mRNA expression of the wild-type and mutant allele and total *KCNQ1* in hiPSC-CMs of Line 1 (*n* = 12). Bottom, allele-specific relative mRNA expression of the wild-type and mutant allele and total *KCNQ1* in hiPSC-CMs of Line 2 (*n* = 6). * *p* < 0.05; ** *p* < 0.025; *** *p* < 0.001 compared to shSCR negative control shRNA; error bars indicate SEM.

**Figure 2 ijms-23-04053-f002:**
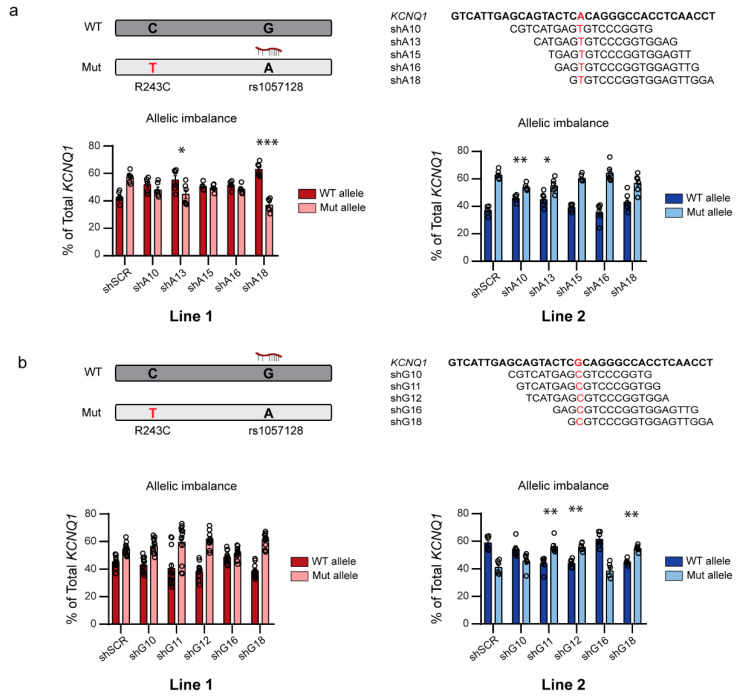
Allelic imbalance induced by allele-specific shRNAs: (**a**) Top, schematic representation of the shRNAs targeting the mutant *KCNQ1* allele in hiPSC-CMs. Bottom, allelic expression of wild-type and mutant *KCNQ1* allele presented as % of total *KCNQ1* expression for Line 1 in red (left; *n* = 6) and Line 2 in blue (right; *n* = 6); (**b**) Top, schematic representation of the shRNAs targeting the wild-type *KCNQ1* allele. Bottom, allelic expression of wild-type and mutant *KCNQ1* allele presented as % of total *KCNQ1* expression for Line 1 in red (left; *n* = 12) and Line 2 in blue (right; *n* = 6). * *p* < 0.05; ** *p* < 0.025; *** *p* < 0.001 compared to allelic expression in the shSCR negative control shRNA; error bars indicate SEM.

**Figure 3 ijms-23-04053-f003:**
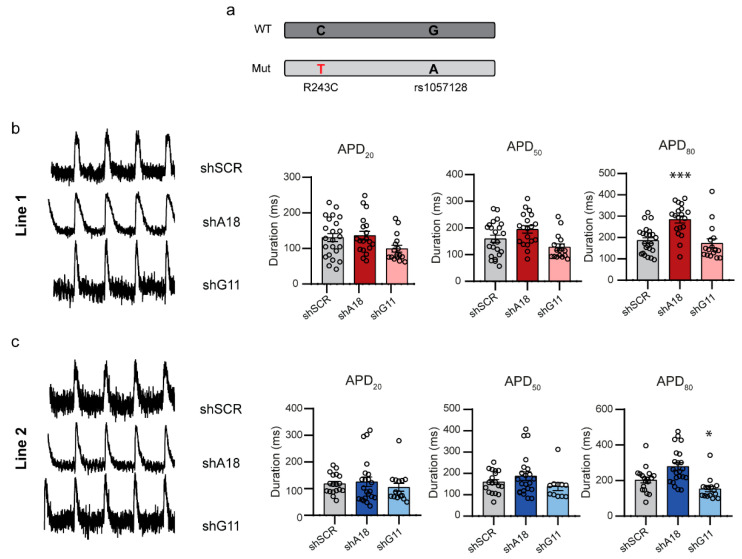
Action potential duration is affected by shifts in allelic balance: (**a**) Schematic representation of the SNP and mutation in *KCNQ1*; (**b**,**c**) Left, typical recordings of optical action potentials derived from ArcLight fluorescence changes in hiPSC-CMs from Line 1 (**b**) or Line 2 (**c**) treated with either negative control shSCR, shA18 targeting the mutant *KCNQ1* allele or shG11 targeting the wild-type *KCNQ1* allele stimulated at 1 Hz. Right, action potential duration at 20, 50, or 80% of repolarization (APD_20_, APD_50_, and APD_80,_ respectively) of optical action potentials of hiPSC-CMs from Line 1 in red (**b**) or Line 2 in blue (**c**). * *p* < 0.05; *** *p* < 0.001 compared to shSCR negative control treated hiPSC-CMs; error bars indicate SEM.

**Figure 4 ijms-23-04053-f004:**
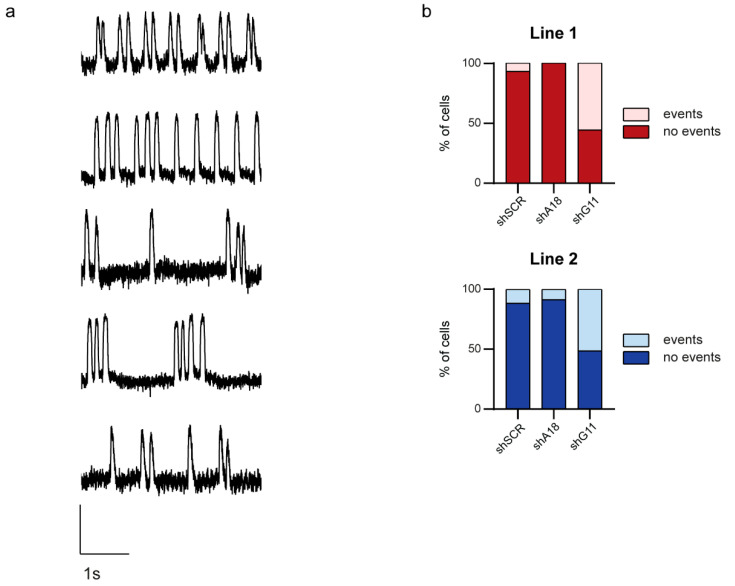
The occurrence of arrhythmic events is affected by allele-specific downregulation of the mutant or wild-type *KCNQ1* allele. (**a**) Typical examples of ArcLight traces showing the fluorescence changes over time of hiPSC-CMs with arrhythmic events; (**b**) Percentage of cells with arrhythmic events in hiPSC-CMs of Line 1 in red (top; shSCR: *n* = 28, 2 with events; shA18: *n* = 19, no events; shG11: *n* = 34, 19 with events) and hiPSC-CMs of Line 2 in blue (bottom; shSCR: *n* = 43, 5 with events; shA18: *n* = 47, 4 with events; shG11: *n* = 41, 21 with events).

**Figure 5 ijms-23-04053-f005:**
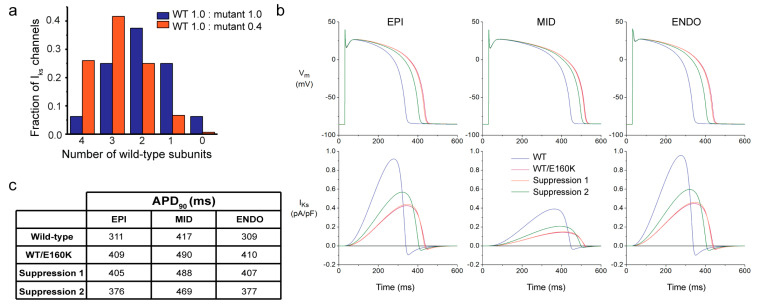
Computer simulation of allele-specific mutant *KCNQ1* inhibition in an adult human cardiomyocyte: (**a**) Fraction of slow delayed rectifier potassium current (I_Ks_) channels with 0-4 wild-type (WT) subunits in case of equal expression of wild-type and mutant KCNQ1 subunits (WT 1.0/mutant 1.0) and with a 60% suppression of mutant KCNQ1 subunits (WT 1.0/mutant 0.4) assuming random co-assembly of subunits into tetrameric channels; (**b**) Effects of changes in I_Ks_ on action potentials of the epicardial (EPI), midmyocardial (MID), and endocardial (ENDO) versions of the human ventricular cell model. Membrane potential (V_m_; top) and associated I_Ks_ (bottom) at 1 Hz stimulation that result from simulations with 100% wild-type *KCNQ1* expression (WT), with an equal heterozygous expression of wild-type and E160K mutant KCNQ1 subunits (WT/E160K), and with a 60% suppression of E160K mutant KCNQ1 subunits, assuming that all channels with mutant subunits contribute equally to mutant I_Ks_ (‘Suppression 1’) or that only channels with one mutant subunit contribute to mutant I_Ks_ (‘Suppression 2’). Note that the red and orange lines largely overlap; (**c**) Values of action potential duration at 90% of repolarization (APD_90_) in each of the simulation settings.

## Data Availability

All data are available within the manuscript or added Appendix A.

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
