# Peer review of "shRNAs Targeting a Common KCNQ1 Variant Could Alleviate Long-QT1 Disease Severity by Inhibiting a Mutant Allele"

_ijms, 2022, doi:10.3390/ijms23074053_

Round 1

Reviewer 1 Report

Clócera-Ortega and co-authors present a very interesting approach for correcting LQT1 through RNA silencing.  They generated two hiPSC-CM models both containing a LQT1-linked R243C variant and then screened numerous shRNAs for their ability to down regulate a specific KCNQ1 allele in these lines. Their approach is creative in designing shRNAs to target a common SNP in the 3’ UTR of the same KCNQ1 allele. This could in theory be used to correct multiple LQT1 variants instead of designing shRNAs for each disease variant.  Indeed, they identified shRNAs that significantly silenced the mutant allele and shifted the allelic balance towards more WT channels, which ultimately reduced the occurrence of arrhythmic events. Finally, they performed computer simulations using three different human ventricular cell models to support this approach beyond their study of just the R243C variant.  The authors’ manuscript is very well written with clear figures and demonstrates a novel approach towards correcting LQT1 variants. However, the claim that this approach alleviates LQT1 disease severity as described in the title is premature and some aspects of the manuscript need to be addressed. 

1. This study does not include control iPS-CMs for comparison with the R243C cell lines. Although, the authors note in the discussion (starting line 372) that the R243C iPS-CMs do not show a long APD compared to other published LQT1 lines, differences between lines should be tested in their lab rather than compared to the literature. The authors should first establish their LQT1-disease model by comparing it's phenotype (e.g. APD) to an isogenic control (ideally) or at a minimum a healthy iPS-CM control.  

2. Along these same lines, Figure 3 shows that shRNA-mediated downregulation of the mutant and WT alleles have the opposite effect on APD than expected. This is a very surprising result and important to mechanistically understand if this is a potential therapy. For example, on line 230, the authors suggest that other channels may be affected, and this is a plausible experimental line to pursue given the trafficking for some other cardiac channels are intertwined (e.g. Nav1.5 and Kir2.1).

3. Also, along these lines, the lactate method used to purify the cardiomyocytes (line 481) was recently shown to have dramatic effects on the CM phenotype and even suggested as a model of ischemic heart failure (PMID: 33878037). As a result, these CMs likely confound some of the results and repeating the data for Figure 3 and 4 using CMS without lactate purification would strengthen this paper. It would also be better for comparisons between control mutant lines and potentially provide insight into the unexpected APD results before pursuing other avenues.

4. The authors only show reduced arrhythmic events for one variant. Since the power of this approach is its potential at correcting numerous variants, at least one more variant with a different loss-of-function mechanism should be studied to support this strategy.

More generally, most of the suggested experiments are fairly involved and perhaps some are beyond the scope of this study. I think the data shown here supports a more toned-down conclusion and the paper could be edited to reflect this. For example, the title could be more specific. Something like: shRNAs targeting a common KCNQ1 variant reduces arrhythmic events in an iPS-CM model.

Minor comments:

1. The R243C variant is first mentioned on line 112 but not described in any detail. It would help to cite any studies describing this variant and its loss of function mechanism (e.g. a quick search found PMID: 10409658).

2. It would also be helpful to provide a rationale for choosing KCNQ1 R243C. As mentioned, LQT1 is effectively treated using beta-blockers so why not choose another example where alternative treatments might be more likely?

3. Along those lines, the novelty of this approach is the ability to correct many variants with just two shRNAs. It might be interesting to comment in the discussion whether other genes might be amenable to this approach (i.e. have common SNPs like KCNQ1 but without good treatments).

4. Similarly, include a discussion point on whether this strategy applies to all types of variants with different loss-of-function mechanisms.

5. It might be helpful to include which cell line is used within the figures instead of just in the legends. For example, Figure 1 middle row could be panel b with “cell line 1” labelled within the figure or y-axis. Same for the following panels an other pertinent figures. 

Reviewer 2 Report

General comments

The study shows generation of a inhibitory RNAs to silence mutations in the KCNQ1 gene by targeting a common single nucleotide polymorphism (SNP), rs1057128, in order to avoid the need to generate siRNAs against every single mutation. Mutations in KCNQ1 are responsible for long-QT syndrome type 1 (LQT1). Patients heterozygous for such mutations co-assemble both mutant and wild-type KCNQ1-encoded subunits into tetrameric Kv7.1 potassium channels, what results in an arrhytmia developing. Designed allele-specific  shRNAs were validated in human induced pluripotent stem cell-derived cardiomyocytes (hiPSC-CMs) from two LQT1 patients. Authors demonstrated that specific inhibition of the mutant KCNQ1 allele prevented the occurrence of arrhythmic events in both analyzed hiPSC-CMs. This approach might help to develop shRNAs improving the LQT1 phenotype in a substantial number of KCNQ1 mutation carriers that are heterozygous for the targeted SNP.

The study appears to have appropriate methodology, the data are clearly presented and the manuscript is well and clearly-written. There are few comments which may be useful:

Figure 1, 2, and 3 - I would recommend to add an information that presented data are obtained from hiPSC-CMs of Line 1 or Line 2 to make these figures more clear.

Round 2

Reviewer 1 Report

The authors gave thorough and cogent responses to each critique and edited the manuscript appropriately throughout. No further edits suggested.